# Exploring the Role of Dietary Calcium Intake in Muscle and Cardiovascular Performance Among Young Athletes

**DOI:** 10.3390/sports12110288

**Published:** 2024-10-22

**Authors:** Pragya Sharma Ghimire, Xiyan Ding, Adam Eckart

**Affiliations:** College of Health Professions and Human Services, Kean University, 1000 Morris Ave, Union, NJ 07083, USA; dingxiy@kean.edu (X.D.); eckarta@kean.edu (A.E.)

**Keywords:** dietary calcium intake, muscle performance, young athletes

## Abstract

The importance of dietary calcium intake in bone metabolism has been well established; however, it is still less investigated in health-related components, especially addressing muscle performance. This study aims to compare dietary calcium intake and its relationship with cardiovascular and muscular performance in young athletes (Lacrosse, Baseball, and soccer players). In this cross-sectional study, 95 participants (ages 18 to 30) participated during two visits to the Human Performance Laboratory. Participants completed body composition, lower and upper body muscle performance, cardiorespiratory protocol, and questionnaires related to dietary calcium intake. One-way ANOVA was used to determine the difference in the calcium intake between groups. The Pearson Correlation Coefficient was used to investigate the relationship between dietary calcium intake and muscle performance variables. Further, linear regression was used to assess the predictive value of calcium variables on overall muscle performance. Lacrosse players had significantly higher calcium intake than Baseball and soccer players (*p* < 0.05). There was a significant positive correlation between dietary calcium intake and the number of push-ups (r = 0.28; *p* = 0.03). CIBW and CI explained 4.3% and nearly 25% of the change in MPI, respectively (*p* < 0.001). This suggests the possible role of dietary calcium intake in enhancing health-related fitness components and highlights the need to explore its involvement in muscle-bone crosstalk.

## 1. Introduction

Mineral and trace elements (MTEs) play indispensable roles in modern healthy diets among young athletes. MTEs such as calcium, magnesium, zinc, and selenium provide a significant role in physiological processes [1] correlated with sports performance, including energy storage/utilization, protein metabolism, inflammation, oxygen transport, cardiac rhythms, bone metabolism, and immune function [2,3]. Additionally, the importance of a balanced diet and engaging in physical activity cannot be emphasized enough for humans. Evidence suggests that dietary calcium intake and exercise focused on enhancing muscle strength may help young people prevent cardiometabolic disease [4].

Calcium is a key mineral for energy metabolism, bone health, muscle contraction, and other physiological processes in the human body. Although the role of Ca^2+^ ions in skeletal muscle function and plasticity is widely accepted, emerging research has drawn attention to exploring the role of calcium in cellular senescence [5]. Calcium-rich foods include dairy products, cereals, nuts, and vegetables, whereas fortified foods such as cereals and juices are additional sources of calcium. It has been reported that 72% of the calcium supply in the United States comes from dairy products [6]. Further, calcium intake has also been linked to play an integral role in bone mineral density and fracture reduction in populations such as children and postmenopausal women [7]. Calcium is the primary component of mineralized tissues, containing 99% of total body calcium, and is pivotal in skeleton mineralization, essential for normal growth, development, and bone strength. The remaining calcium is distributed within the blood, extracellular fluids, muscles, and tissues [8,9]. In these locations, calcium facilitates muscle contraction, vascular contraction and vasodilation, nerve impulse transmission, and intra and extracellular signaling [9,10].

It has been widely accepted that macronutrients play an important role in using muscle fuel and providing support to meet specific exercise demands [11]. It is worth noting that athletes’ physical activity, performance, and micronutrient consumption must align with adequate standards [12]. Meeting the daily requirements seems crucial in addressing the deficiency related to specific MTEs among athletes for those who cannot meet their daily requirements. In this context, it is imperative to understand that athletes participating in competitive sports may encounter the challenges associated with injury risks necessitating prolonged recovery periods and adherence to a balanced diet. While no single dietary approach has been universally proven to optimize performance, emerging research has highlighted the Mediterranean diet as a model for enhancing athletic performance and overall well-being among athletes [13]. It is important to note that diet and training could impact athletic performance and overall health due to the extreme cardiovascular, metabolic, and energy demands. Further, due to inadequate dietary intake, changes in body composition may negatively affect athletic performance during the competitive season.

Irrefutable evidence supports vitamin D regulates calcium absorption and mineralization while maintaining muscle mass, strength, and overall physical performance. This is particularly evident in older individuals, highlighting the role of vitamin D and calcium in muscle performance and mitigating sarcopenia [14], especially in older people, suggesting its importance in muscle performance. However, no direct evidence supports calcium supplementation’s efficacy in athletic performance [1]. It is worth noting that calcium’s role in muscle contraction can mitigate the impact of increased levels of parathyroid hormones, which stimulate bone resorption [15] and increase the risk of osteoporosis. In this sense, it is important to acknowledge that a research gap exists in exploring the role of dietary calcium intake and muscle performance in young athletes.

Cardiorespiratory fitness (CRF) and both upper and lower body performance assessment are important parameters in developing training programs among athletes. The assessment of VO_2_ max has been recommended by The American College of Sports Medicine and the American Heart Association in clinical and sports settings [16]. Furthermore, dietary calcium intake and calcium supplementation’s role in preventing cardiovascular disease (CVD) prevention remains controversial. Systematic and meta-analysis studies reported that higher calcium intake is not associated with CVD [17]. In contrast, another study reported a potential association between higher calcium intake and an increased mortality risk [18]. It is worth noting that the population samples in both studies did not include athletes.

While the role of dietary calcium intake in bone metabolism is well established, a wide research gap remains in understanding the relationship between calcium intake and muscle performance, especially in young athletes. Therefore, the present study aims to compare dietary calcium intake across different sports and its relationship with cardiovascular and muscular performance in young athletes. We hypothesized that there would be a variation in dietary intake between soccer players, lacrosse players, and baseball players and that dietary calcium levels would be associated with cardiovascular and muscle performance.

## 2. Materials and Methods

This non-randomized cross-sectional design included three independent groups (Baseball, Soccer, and lacrosse) of male athletes 18–30 years old. The available data on calcium intake and its relationship with health-related components are sparse. Calcium intake results from a cross-sectional study by Torres-Costoso et al., (2021) were used to perform a power analysis using G*Power 3.1 for 80% Power based on α = 0.05 [4,19]. The effect size (d) for calcium intake differences between men and women is determined to be 0.72, a medium effect size [20]. Therefore, a total of 98 young male athletes were screened for their eligibility. A total of 3 participants were lost in follow-up and were excluded. Participants were categorized into three categories (Baseball, Lacrosse, and Soccer) based on their current involvement in sports. Baseball (n = 49), Soccer (n = 33), and Lacrosse (n = 13) athletes from diverse ethnicities were included in this study. Participants in this study were screened for inclusions and exclusions criteria. All the participants in this study were healthy and recreationally active. Participants without cardiovascular and metabolic diseases or physical disabilities were included in this study. Participants who met the criteria of not taking any medications that affect muscle and metabolism were included in this study. Prior to the study, all the participants were provided with details of the study protocol outlining the risks and benefits of the study. Participants provided the written informed consent form before participating in this study. All the protocols were approved by the Kean University Institutional Review Board # 22-112114.

### 2.1. Protocol

In this study, participants completed two visits to the Human Performance Laboratory. Participants completed two visits to the Human Performance Laboratory. During the first visit, participants completed a written consent form, a health history questionnaire, a Bone-specific Physical Activity Questionnaire (BPAQ) [21], and a Physical Activity Readiness Questionnaire (PAR-Q) and familiarized themselves with the testing protocol.

Participants were instructed to complete a dietary calcium intake questionnaire. This questionnaire assessed daily calcium intake by gathering information on the frequency of specific foods consumed daily or weekly. Additionally, data on calcium supplements, including types, dosage, and generic names, were recorded at the end of the questionnaire [22]. Details of the protocol are provided in Figure 1.

### 2.2. Body Composition Variables

Prior to the muscular performance assessment, each participant’s height and weight were measured using a stadiometer (Novel Products, Rockton, IL, USA) and a digital scales bioimpedance analysis system (Tanita Inc., Arlington Heights, IL, USA), respectively. Furthermore, variables such as %body fat, fat-free mass, fat mass, and total body water were collected.

### 2.3. Muscle Performance Variables

During the second visit, each participant’s lower body muscular strength was measured using the vertical jump test (Just Jump System, Perform Better, Warren, PA, USA) with a Tendo FitroDyne (Tendo Sports Machines, Trencin, Slovak Republic). Participants’ upper body strength was measured by handgrip test using handgrip dynamometry (Takei Scientific Instruments, Yashiroda, Niigata, Japan). Participants were instructed to perform three countermovement jumps, each separated by a minute. The three trials were recorded and were averaged for the analysis. Muscular power was estimated using force and velocity data from the Tendo machine, and relative power was calculated by dividing the average muscular power by each participant’s body weight. Participants were instructed to flex their elbows 0–30 degrees dorsiflexion and 0–15 degrees ulnar deviation in both their dominant and non-dominant hands. Participants’ grip width was adjusted, and they were instructed to squeeze as hard as possible for 3–5 s. Each participant completed three trials on each side, with one minute of rest between the trials. The highest recorded maximal grip strength from both hands was included in the data analyses.

Participants’ upper body muscular endurance was assessed by performing the push-up test. Participants were asked to remain in the standard down position with their hands pointing forward and under their shoulders, back straight, and head up, using their toes as the pivotal point. Participants were instructed to raise their bodies by straightening their elbows and to return to the down position until the chin touched the mat. The participants were asked to keep their backs straight and push up to a straight arm position. The maximal number of push-ups performed consecutively without rest for a minute counted as their score. The test was terminated if the participant strained forcibly or could not maintain the appropriate technique within two repetitions. The variables were analyzed, including time in the air, jump height, jump velocity, and power.

### 2.4. Cardiorespiratory Fitness Test

Using the Bruce Protocol, participants’ cardiorespiratory fitness assessment (CRF) was measured by a graded treadmill exercise test (Trackmaster, Newton, KS, USA). Participants were asked to perform the graded exercise protocol until exhaustion. The graded exercise test comprised seven steps with different treadmill settings, which started at 1.7 mph speed with a 10% gradient. The speed and gradient of the treadmill were elevated every 3 min. The test was terminated if the participant reached volitional fatigue, could not continue exercise, or requested test termination. The participant’s resting and post-exercise hemodynamic responses (heart rate and blood pressure) were also measured during this test.

### 2.5. Statistical Analysis

Data were analyzed using IBM SPSS 27.0 (SPSS Inc., Chicago, IL, USA). All descriptive statistics are reported as mean ± standard deviation (SD) unless otherwise stated. All dependent variables were tested for normality using the Kolmogorov-Smirnov test. The One-way ANOVA was used to determine the effects of calcium intake on body composition, muscle performance, heart rate, blood pressure, and VO_2_ max in three groups. Pearson Product Moment Correlation coefficients were used to estimate the relationships between calcium intake and neuromuscular performance variables. Spearman Rho correlation was used for non-normally distributed data. We also included BPAQ scores and dietary calcium intake in predicting cardiorespiratory fitness variables using stepwise regression models. After estimating total calcium intake, the ratio of calcium intake to body weight (CIBW) (lbs.) was calculated for each participant to account for individual differences in total energy intake. First, we explored the associations of CIBW using Pearson product-moment correlation, and then, linear regression was used to assess the predictive value of calcium variables on overall muscle performance using an index score. Muscle performance index (MPI) scores were computed to evaluate the associations between calcium intake and overall muscle performance. Scores for each muscle performance variable were standardized (percentile rank) and then averaged. Participant characteristics and individual muscle performance variables were imputed into a linear regression model to evaluate the influence of each variable on overall muscle performance. The level of significance was set at *p* ≤ 0.05.

## 3. Results

Table 1 shows a significant difference in weight and BMI between the three groups, where lacrosse players had significantly higher calcium intake compared to Baseball and soccer players (*p* < 0.05). We also found that soccer players had significantly higher current cBPAQ scores than baseball players (*p* < 0.01). In contrast, baseball players had significantly higher pBPAQ scores than baseball players (*p* < 0.01). There was no significant difference observed in the body composition variables, including BMR, body fat percentage, fat mass, fat-free mass, and TBW (*p* < 0.05) (Table 2).

We found that baseball players had significantly higher time in air and vertical jump height compared to soccer and lacrosse players (*p* < 0.01). In contrast, lacrosse players had significantly higher relative power (*p* < 0.05) (Table 3). There was also a significant difference in right and left-hand grip strength, suggesting that baseball and lacrosse players had significantly higher left and riTheght-hand grip strength than soccer players (*p* < 0.01).

Table 4 shows that lacrosse players had a significantly higher number of push-ups and post-exercise systolic blood pressure compared to Baseball and soccer players (*p* < 0.01). VO_2_ max was not significantly different between groups (*p* > 0.05).

A significant positive correlation was found between dietary calcium intake and the number of push-ups (Figure 2). Surprisingly, we also found a significant negative correlation between dietary calcium intake with VO_2_ max and post-exercise heart rate, respectively (Figure 3 and Figure 4). We found no significant association between dietary calcium intake and lower and upper-body muscle performance variables (*p* > 0.05). No significant correlations were observed between dietary calcium intake and body composition variables (*p* > 0.05). We also found no significant difference between dietary calcium intake and resting and post-exercise blood pressure variables (*p* > 0.05). Similarly, no significant differences were found between dietary calcium intake and resting heart rate (r = 0.58; *p* = 0.67). Further, there was a trend (*p* = 0.06) in determining the relationship between dietary calcium intake and total and past BPAQ scores.

Figure 5 shows a mean rank for the muscle performance index based on the sports. We found that the muscle performance index was significantly higher in baseball players compared to soccer and lacrosse players (*p* < 0.05).

We also included BPAQ scores and dietary calcium intake in predicting cardiorespiratory fitness variables using stepwise regression models (Table 5). We performed stepwise regression analysis using tBPAQ, cBPAQ, pBPAQ, and calcium intake scores to predict muscle performance and cardiorespiratory variables. We found that cBPAQ scores predict time in air, vertical jump, relative power, and hand grip strength, respectively, 6%, 7%, 4%, and 12%. We also found that calcium intake accounted for 7% and 10% of the number of push-ups and VO_2_ max, respectively.

Table 6 shows several individual and muscle performance variables were significantly inversely correlated with CIBW, including weight (−0.273), BMI (−0.202), power (−0.252), post-exercise heart rate (−0.286), and VO_2_ max (−0.329). However, variables such as push-ups (0.291) and tBPAQ (0.212) were positively correlated with CIBW.

Table 7 shows that CIBW and CI explained 4.3% and nearly 25% of the change in MPI, respectively (*p* < 0.001).

Table 8 shows displays a hierarchal multiple regression model predicting MPI. In Model 1, TIA (ΔR^2^
*=* 44.6%) had the most influence on MPI, followed by power (ΔR^2^
*=* 27.8%), velocity (ΔR^2^
*=* 9.0%), LHG (ΔR^2^
*=* 6.5%), and VO^2^ (ΔR^2^
*=* 4.5%). All predictors explained 92.4% of the change in MPI (*p* < 0.001).

## 4. Discussion

The major findings from this study indicate that dietary calcium intake is significantly different between lacrosse and soccer players. We also found that BMI is significantly different between soccer and baseball players. Both upper and lower body muscle performance showed significant differences across the groups. Furthermore, we found that dietary calcium intake is positively related to the number of push-ups and negatively related to aerobic measures and post-exercise heart rate. In the current study, soccer players had the lowest dietary calcium intake among the three sports.

Irrefutable evidence supports that athletic performance relies on adequate nutrition, training, and overall health. In addition to macronutrients, athletes should focus on high-quality protein, vitamins, and minerals. Dietary calcium intake is often regarded as integral to bone health, muscle contraction, and other physiological processes in the human body [23]. Although no evidence supports that calcium supplementation directly affects athletic performance [1], in contrast, one study has shown that calcium supplementation might benefit athlete’s dietary deficiencies [24]. It is evident that calcium plays an important role in muscle contraction and can mitigate the impact of increased levels of parathyroid hormones, which stimulate bone resorption [15]. It is worth noting that lacrosse, Baseball, and Soccer each rely on different energy systems. Compared to lacrosse and Baseball, soccer places significant demands on aerobic and anaerobic systems along with longer recovery time. The lower dietary calcium intake observed among soccer players in this study may be attributed to the endurance-oriented nature of the sport and calcium loss experienced by the athletes during strenuous activity. Supporting this, Galanti et al. [25] found low levels of calcium intake in a sample of young endurance athletes. Notably, all participants in this study on the soccer team were in-season during the data collection process, aligning with the findings from the previous study by [26], which suggest that strenuous activity can reduce serum phosphate and ionized calcium levels and elevate vitamin D metabolites in athletes. However, it is also important to note that this study did not measure any serum markers of bone and muscle metabolism.

Various studies have compared the effects of body composition on performance in football, soccer, and lacrosse and have supported that fat mass and percent body fat are different between the sports and exhibit different roles to be considered successful [27,28,29]. Further, sports that require speed, power, muscular strength, and high force production are recognized as key determinants of athletic success. In this study, lower and upper body muscle performance was significantly higher in baseball and lacrosse players than in soccer players. Strength is a significant predictor of overall performance, and elite players across different sports contexts have shown markedly superior handgrip scores compared to non-elite counterparts [30]. In the current study, LHG strength was an important predictor of overall muscle performance as measured by the MPI. Hand dominance was not collected; however, it can not necessarily be inferred from right versus left-hand grip performances. Notwithstanding, baseball players had the smallest difference between right- and left-hand grip scores. Evidence suggests left-handed throwers have minimal limb asymmetries with greater musculature and strength in both upper limbs compared to their right-handed counterparts [31]. It is possible that the link between LHG and MPI is driven by better performances from left-handed throwers, specifically in the baseball group. Although this study highlighted the predictive value of muscle power (TIA, velocity, and power) and muscle mass variables (TBW and FFM) on overall muscle performance, VO_2_ max was also a strong predictor of overall muscle performance (MPI), favoring soccer players in MPI compared to lacrosse players.

A meta-analysis study reported that countermovement jump height is a valid marker for neuromuscular fatigue and has often been employed in routine test batteries, with many sports performance actions occurring unilaterally, especially in soccer players [32]. Several studies reported a decrease in vertical jump performance towards the end of the season compared to preseason, which is in line with this study as we collected the data towards the end of the in-season among soccer players [33]. The vertical jump height and power observed in this study were higher than in previous studies among soccer and lacrosse players [34,35], and it is worth noting that participants utilized their arms during the countermovement jump, similar to the approach in this study.

This study also found significantly higher post-exercise systolic blood pressure in baseball players compared to lacrosse and soccer players. It is well established that determinants of hemodynamic responses during exercise depend on age, sex, cardiovascular fitness, body fat, medication, and participation in different types of exercises. Although this study did not find significant differences in aerobic fitness measures, including VO_2_ max, the higher post-exercise systolic blood pressure response among baseball players warrants further investigation. Insufficient evidence exists to define what constitutes a normal systolic blood pressure response to exercise in trained individuals. There is also a lack of evidence linking elevated systolic blood pressure during graded exercise testing to pathology. Further, as our knowledge expands, there is an increased agreement that monitoring post-exercise blood pressure during graded exercise testing could emerge as a vital tool in foreseeing early cardiovascular events, particularly in athletes [36].

In Spearman’s Rho correlation analysis, we found a significant inverse relationship between dietary calcium intake and VO_2_ max suggesting the role of calcium intake not only for bone health but also for exercise capacity, which is in contrast to the previous finding by [37], where no relationship was observed on exercise capacity. However, few studies have explored the role of dietary calcium intake on exercise capacity, with the majority focusing on commercial calcium supplementation rather than dietary intake sources. Additionally, while evidence suggests that calcium supplementation modulates adiposity and weight management, this study found a negative correlation between calcium intake, body weight, and BMI, which aligns with the previous findings by [4,38]. A study conducted by Zemel et al. [38] proposed a plausible mechanism that using an animal model that found intracellular calcium levels could impact adipocyte fat metabolism via reduced 1,25-vitamin D levels, leading to decreased calcium influx into cells and lowered intracellular ion levels, ultimately stimulating lipolysis and inhibiting lipogenesis. Conversely, high intracellular calcium levels (resulting from low dietary calcium) were associated with increased fat synthesis and reduced lipolysis via calcium-dependent mechanisms. However, these studies did not provide a definitive answer but prompted further research. Additionally, dietary calcium could impact fat metabolism by modulating adipocytes and increasing fecal fat excretion, especially during a high-calcium diet in the short term. Therefore, understanding the triad of body composition, exercise performance, and dietary calcium intake warrants further investigation.

While exploring this intricate link between muscle strength and dietary calcium intake, surprisingly, we found that calcium intake and body weight variables significantly predicted muscle performance index, underscoring its important role in muscle function. Although there were no significant differences, there was a trend toward lower CIBW in baseball and soccer players compared to lacrosse players (*p* = 0.058) despite better MPI scores in Baseball and soccer players. These findings might reflect calcium intakes that do not match the demands of high-performing young athletes [23]. Alternatively, these findings could be confounded by differences in in-season versus off-season behaviors across sports. Soccer was in season during data collection, while baseball and lacrosse were off-season. Overall workload-to-recovery ratios during in-season and off-season training/practice probably vary within and between each sport. For example, evidence suggests that energy expenditures and physical activity levels differ between collegiate lacrosse players and basketball players during team-based activities [39]. Furthermore, higher BMI and lower overall muscle performance (MPI) in lacrosse players compared to soccer and baseball players supports lower efflux of calcium, resulting in downstream effects on body composition. More research is needed, however, to clarify the effects these differences have on muscle performance, body composition, and dietary patterns.

While the role of sarcopenia in modulating calpains, which regulates myogenesis, has been studied, other studies have shown no significant differences between calcium intake and sarcopenia in older populations [40,41,42]. It is worth noting that participants in the current study were young and healthy adults, unlike those in previous studies [40,41]. Therefore, the mechanisms behind calcium intake and muscle performance warrant further investigation. Interestingly, we also found that BPAQ scores are a predictor of muscle performance variables. Although BPAQ scores were widely reported as a predictor of bone mineral density [21,43], further investigation is warranted to measure possible muscle and bone crosstalk along with calcium intake.

## 5. Conclusions

This study is subject to certain limitations, as our findings need to be interpreted within a cross-sectional research design context. This study is based on self-reported dietary calcium intake questionnaires that may be deemed subjective. The reported correlations between dietary calcium intake and muscle performance variables do not explain the causality. Further, we did not measure participants’ bone mineral density and serum calcium levels, which could have provided some insights to enhance our understanding of this area.

It is important to note that dietary planning plays a crucial role in determining athletes’ physical and cognitive performance, yet micronutrients are often overlooked compared to macronutrients in their diet. Poor micronutrient composition could possibly be due to a lack of education, while sports nutrition knowledge has significantly proven appropriate food selection among athletes. Therefore, future research suggestions should incorporate wide-ranging sports teams along with intervention studies in male and female athletes and further expand research on cultural differences in diet and pattern of diet along with intervention studies.

## Figures and Tables

**Figure 1 sports-12-00288-f001:**
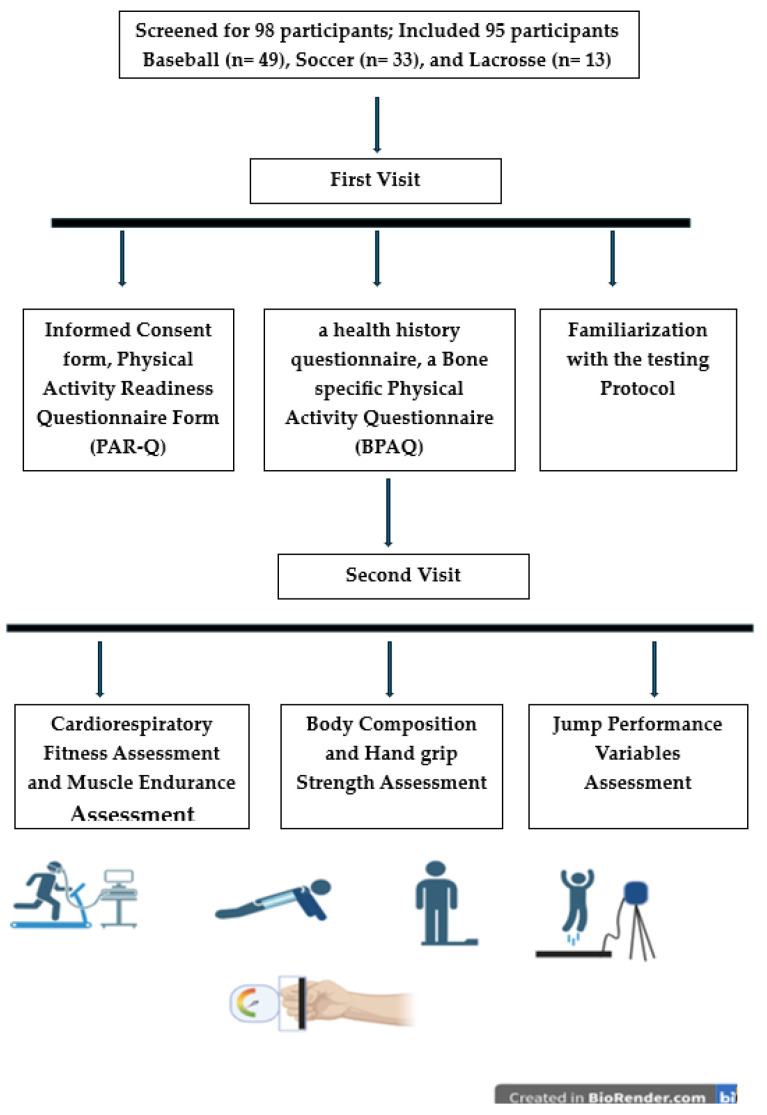
Research Protocol time.

**Figure 2 sports-12-00288-f002:**
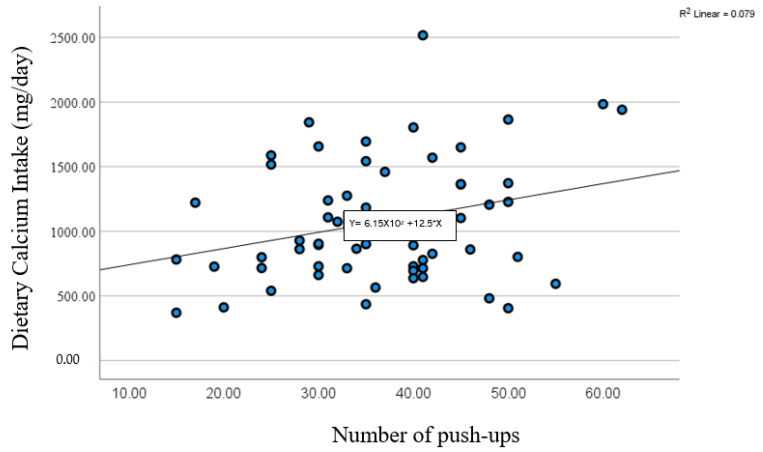
A significant positive correlation between dietary calcium intake and the number of push-ups (r = 0.28; *p* = 0.03) (n = 57).

**Figure 3 sports-12-00288-f003:**
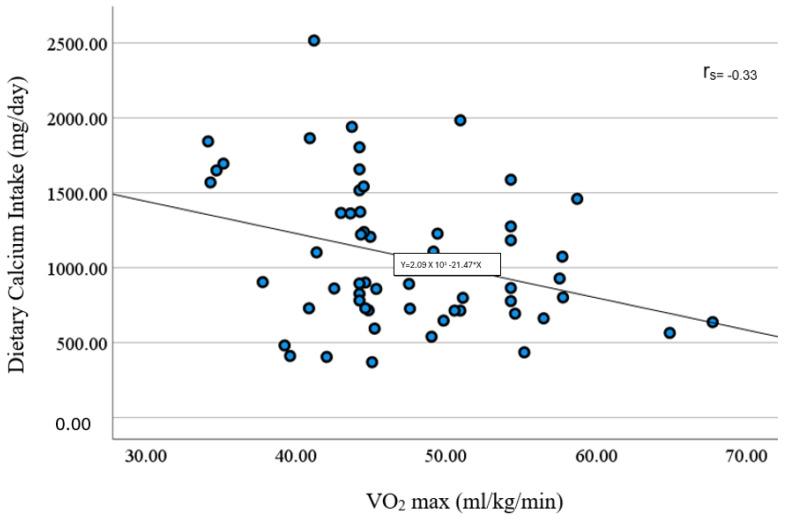
A significant negative correlation between dietary calcium intake and VO_2_ max (r_s_ = −0.33; *p* = 0.01) (n = 57).

**Figure 4 sports-12-00288-f004:**
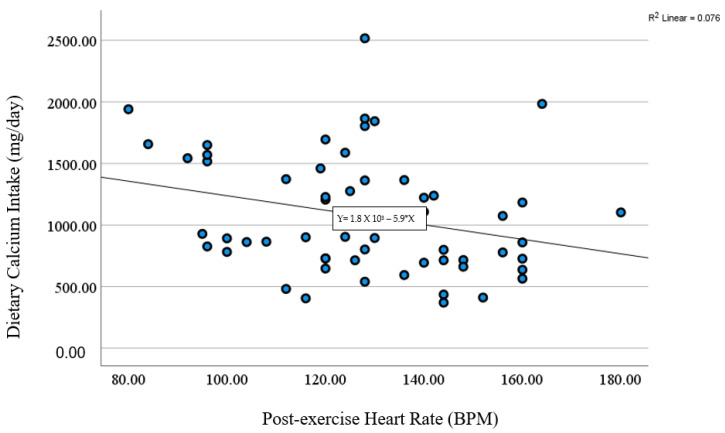
A significant negative correlation between dietary calcium intake and post-exercise heart rate (r = −0.27; *p* = 0.03) (n = 57).

**Figure 5 sports-12-00288-f005:**
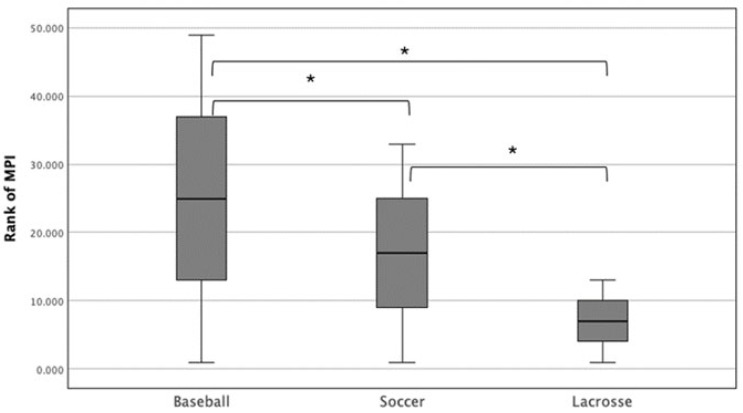
A mean rank for muscle performance index based on the sports shows that the muscle performance index was significantly higher in baseball players compared to soccer and lacrosse players. * significant *p* < 0.05.

**Table 1 sports-12-00288-t001:** Physical Characteristics, BMI, Calcium Intake, and Bone-Specific Physical Activity Scores of Participants (Mean ± SD).

Variables	Baseball Players (n = 49)	Soccer Players (n = 33)	Lacrosse Players (n = 13)
Age	19.89 ± 1.37	19.96 ± 1.59	19.46 ± 2.43
Height (cm)	180.10 ± 7.25	176.92 ± 7.90	177.80 ± 6.47
Weight * (kg)	83.37 ± 11.52	74.68 ± 9.32	82.11 ± 9.67
BMI *	25.71 ± 3.52	23.89 ± 3.06	25.91 ± 2.35
Calcium Intake (mg/day) *	1091.86 ± 437.91	928.64 ± 309.61	1321.25 ± 540.06
Calcium to body weight ratio	12.40 ± 5.32	12.70 ± 4.60	16.34 ± 6.75
cBPAQ **	7.77 ± 3.04	16.26 ± 14.93	10.75 ± 4.08
pBPAQ **	51.68 ± 21.08	49.12 ± 30.25	70.44 ± 34.24
tBPAQ *	29.86 ± 10.89	37.08 ± 18.79	40.59 ± 16.96

BMI: Body Mass Index; cBPAQ: Current Bone-specific Physical Activity Questionnaire; pBPAQ: Past Bone-specific Physical Activity Questionnaire; tBPAQ: Total Bone-specific Physical Activity Questionnaire. * significant *p* < 0.05; ** significant *p* < 0.01.

**Table 2 sports-12-00288-t002:** Body Composition Variables of Participants (Mean ± SD).

Body Composition	Baseball Players (n = 30)	Soccer Players (n = 14)	Lacrosse Players (n = 13)
BMR (kcal)	1950.53 ± 156.78	2712.07 ± 2076.04	1958.46 ± 151.82
% Body Fat	12.11 ± 3.79	13.64 ± 7.83	12.06 ± 4.76
Fat mass (Kg)	22.05 ± 8.83	24.61 ± 16.84	22.50 ± 11.83
Fat-Free Mass (Kg)	154.90 ± 15.45	147.18 ± 15.86	158.15 ± 12.52
TBW (Kg)	110.01 ± 18.64	107.75 ± 11.61	115.77 ± 9.18

BMR: Basal Metabolic Rate; TBW: Total Body Water.

**Table 3 sports-12-00288-t003:** Lower and Upper Body Muscle Performance Variables of Participants (Mean ± SD).

Lower Body Muscle Performance	Baseball Players (n = 49)	Soccer Players (n = 33)	Lacrosse Players (n = 13)
TIA (s) **	0.71 ± 0.04	0.64 ± 0.05	0.65 ± 0.01
VJ height (inch) **	24.93 ± 2.92	20.58 ± 3.59	21.32 ± 1.24
Velocity	1.47 ± 0.18	1.39 ± 0.14	1.42 ± 0.06
Power (Watt)	1000.82 ± 367.23	1012.35 ± 184.99	1130.89 ± 142.89
Relative Power * (kg/Watts)	11.88 ± 3.81	13.52 ± 1.56	13.77 ± 0.74
RHG **	48.64 ± 9.83	40.15 ± 8.95	47.52 ± 8.05
LHG **	47.93 ± 10.32	39.02 ± 8.52	48.69 ± 7.27

TIA: Time in the air; VJ: Vertical Jump; RHG: Right-hand grip; LHG: Left-hand grip; * significant *p* < 0.05; ** significant *p* < 0.01.

**Table 4 sports-12-00288-t004:** Cardiorespiratory Fitness Variables and Push-up Numbers of Participants Mean ± SD.

Cardiorespiratory Fitness	Baseball Players (n = 30)	Soccer Players (n = 14)	Lacrosse Players (n = 13)
Number of Push-up **	37.36 ± 8.87	28.57 ± 8.41	43.30 ± 12.06
RHR	67.26 ± 6.63	71.23 ± 6.28	74.61 ± 25.12
PEBPD (mmHg)	72.20 ± 5.31	72.85 ± 5.64	72.46 ± 3.47
PEHR	122.83 ± 20.61	138.71 ± 18.81	127.38 ± 27.51
RBPS (mmHg)	111.50 ± 9.19	112.42 ± 10.81	113.23 ± 7.32
RBPD (mmHg)	71.13 ± 5.08	71.28 ± 4.68	70.46 ± 5.17
PEBPS ** (mmHg)	164.93 ± 20.98	143.28 ± 17.46	168.00 ± 19.14
VO_2_ max (mL/kg/min)	47.64 ± 6.85	49.38 ± 8.62	43.70 ± 5.61

RHR: Resting heart rate; PEHR: Post-exercise heart rate; RBPS: Resting blood pressure systolic; RBPD: Resting blood pressure diastolic; PEBPS: Post-exercise blood pressure systolic; PEBPD: Post-exercise blood pressure diastolic; ** significant *p* < 0.01.

**Table 5 sports-12-00288-t005:** Regression analysis results for muscle performance and cardiorespiratory Variables.

Dependent Variables	Predictor Variables	β	SEE	R^2^	*p*
TIA (s) **	cBPAQ	−0.001	0.00	0.06	<0.01
VJ (inches) **	cBPAQ	−0.10	0.03	0.07	<0.01
RP (Watts/kg) *	cBPAQ	0.06	0.03	0.04	<0.04
RHG (kg) **	cBPAQ	−0.38	0.10	0.12	<0.01
LHG (kg) **	cBPAQ	−0.38	0.10	0.12	<0.01
Number of Push-ups *	CI	0.006	0.00	0.07	<0.03
VO_2_ max (mL/kg/min) **	CI	−0.005	0.02	0.10	<0.01

TIA: Time in the air; VJ: Vertical Jump; RP: Relative power; RHG: Right-hand grip; LHG: Left-hand grip; cBPAQ: current Bone Physical Activity Questionnaire; CI: Calcium Intake; β-Standardized Regression Coefficient; * *p* < 0.05; ** *p* < 0.01.

**Table 6 sports-12-00288-t006:** Correlations between CI:BW Ratio and Performance/Individual Characteristics.

	Calcium: BW Ratio	
	R	Sig. (2-Tailed)
Weight	−0.273 **	0.007
Body Mass Index	−0.202 *	0.049
Body fat	−0.146	0.277
Basal Metabolic Rate	0.111	0.410
Fat Free Mass	−0.123	0.363
Fat Mass	−0.175	0.193
Total Body Water	−0.008	0.951
Time In Air	−0.167	0.107
Age	−0.103	0.322
Vertical Jump	−0.130	0.210
Velocity	−0.015	0.883
Power	−0.252 *	0.014
Relative power	−0.142	0.170
Right Hand Grip	−0.120	0.245
Left Hand Grip	−0.160	0.123
PUSH-UP	0.291 *	0.028
Post Exercise Heart Rate	−0.286 *	0.031
VO_2_ max	−0.329 *	0.012
tBPAQ	0.212 *	0.040

tBPAQ: Total Bone-specific Physical Activity Questionnaire; ** Correlation is significant at the 0.01 level; * Correlation is significant at the 0.05 level.

**Table 7 sports-12-00288-t007:** Prediction Models for Muscle Performance Index (Calcium).

Model	R	R Square	S.E.	Change Statistics	
				R Square Change	Sig. F Change
1	0.207	0.043	18.03856	0.043	<0.001
2	0.498	0.248	15.99088	0.205	<0.001

Dependent Variables: average percent rank of muscle performance index (MPI); CI: Calcium Intake. 1. Predictors: (Constant), calcium intake relative to body weight. 2. Predictors: (Constant), calcium intake relative to body weight, CI.

**Table 8 sports-12-00288-t008:** Hierarchical Multiple Regression Predicting MPI.

Model	R	R²	Adjusted R²	S.E.	R² Change	F Change	Sig. F Change
1	0.668	0.446	0.436	10.90	0.446	44.20	<0.001
2	0.850	0.723	0.713	7.77	0.278	54.21	<0.001
3	0.902	0.813	0.802	6.45	0.090	25.37	<0.001
4	0.937	0.877	0.868	5.27	0.065	27.43	<0.001
5	0.961	0.923	0.915	4.23	0.045	29.73	<0.001

Dependent Variable: average percent rank of muscle performance tests. TIA: Time in Air; LHG: Left-hand Grip; VO_2_: Maximal Oxygen Uptake (VO_2_). 1. Predictors: (Constant), TIA. 2. Predictors: (Constant), TIA, Power. 3. Predictors: (Constant), TIA, Power, Velocity. 4. Predictors: (Constant), TIA, Power, Velocity, LHG. 5. Predictors: (Constant), TIA, Power, Velocity, LHG, VO_2_.

## Data Availability

The raw data supporting the conclusions of this article will be made available by the authors upon request.

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
