# Peer review of "Exploring the Role of Dietary Calcium Intake in Muscle and Cardiovascular Performance Among Young Athletes"

_sports, 2024, doi:10.3390/sports12110288_

Round 1

Reviewer 1 Report

Comments and Suggestions for Authors

Dear Authors thank you for the manuscript sent for review, after analysis I suggest to make corrections within certain sections:

Material and methods:

-please add in this section a graph showing the course of the research experiment, including the various stages, the size of the research group and the criteria for inclusion and exclusion from the study

- please consider also adding in this section (procedure) pictures of graphs illustrating the course of research and tests, which will be an attractive form of communication for the recipient.

Results:

-please highlight the significantly static results in the tables and provide the units, for example in Table 2 body fat?, fat mass?etc.

In your paper, please also add a limitations section and describe what the limitations of the study were, what the strengths and weaknesses of the experiment were, as well as visions and plans for expansion and future research. 

Author Response

-please add in this section a graph showing the course of the research experiment, including the various stages, the size of the research group and the criteria for inclusion and exclusion from the study

- please consider also adding in this section (procedure) pictures of graphs illustrating the course of research and tests, which will be an attractive form of communication for the recipient.

 Thank you for your suggestions, we incorporated your feedback:

All participants who were healthy and recreationally active, without cardiovascular and metabolic diseases or physical disabilities, and not taking any medications that affect muscle metabolism were included in this study. Participants were screened for the above-mentioned inclusion/exclusion criteria prior to their first visit. We also included the visual representation of research protocol.

Results:

-please highlight the significantly static results in the tables and provide the units, for example in Table 2 body fat?, fat mass?etc.

Thank you for your feedback. We added the units of measured variables.

 Table 2. Body Composition Variables of Participants (Mean ± SD)

Body Composition

Baseball Players   (n=30)

Soccer Players

(n=14)

Lacrosse Players (n=13)

BMR (kcal)

1950.53 ± 156.78

2712.07 ± 2076.04

1958.46 ± 151.82

% Body Fat

    12.11 ± 3.79

    13.64 ± 7.83

    12.06 ± 4.76

Fat mass (Kg)

    22.05 ± 8.83

    24.61 ± 16.84

    22.50 ± 11.83

Fat-Free Mass (Kg)

  154.90 ± 15.45

  147.18 ± 15.86

   158.15 ± 12.52

TBW (Kg)

  110.01 ± 18.64

  107.75 ± 11.61

  115.77 ± 9.18

BMR; Basal Metabolic Rate; TBW; Total Body Water;

In your paper, please also add a limitations section and describe what the limitations of the study were, what the strengths and weaknesses of the experiment were, as well as visions and plans for expansion and future research. 

Thank you for your valuable feedback. We have included the following information

It is important to note that dietary planning plays a crucial role in determining athletes' physical and cognitive performance, yet micronutrients are often overlooked compared to macronutrients in their diet. Poor micronutrient composition could possibly be due to a lack of education, while sports nutrition knowledge has significantly proven appropriate food selection among athletes. Therefore, future research suggestions should incorporate wide-ranging sports teams along with intervention studies in male and female athletes. Further, future studies should also expand research on cultural differences in diet and patterns of diet along with intervention studies.”

Reviewer 2 Report

Comments and Suggestions for Authors

This study explores the relationship between dietary calcium intake and muscle performance in young athletes. The study implies that improving dietary calcium intake might have broader benefits beyond bone health, particularly in athletic performance and overall musculoskeletal health, making it an important factor to consider in dietary interventions for athletes or individuals aiming to optimize physical performance.
This is well written article and the topic is of great significance. The research was conducted according to the principles and rules that apply to cross-sectional examination.
The methodology is thorough and facilitates an analysis of the research data.

However, the manuscript has the following deficiencies
1. The title of the paper "Exploring The Role of Dietary Calcium Intake In Muscle Performance Among Young Athletes" is not aligned with the objective of the study (as mentioned in the abstract, L9-10, and similarly in the research goal stated in the Introduction section, L77-79), where it is stated, "This study aims to compare dietary calcium intake and its relationship with cardiovascular and muscular performance in young athletes (lacrosse, baseball, and soccer players)," which introduces the term "cardiovascular performance." Why is the term "Cardiovascular Performance" not included in the title, but only "Muscle Performance" is?;

2. In the section Materials and Methods, specifically subsection 2.2 Body Composition Variables, the testing protocol is described, but there is no mention of the variables. The same applies to subsection 2.3 Muscle Performance Variables, where the protocol is described, but the variables are not mentioned. Finally, in subsection 2.4 Cardiorespiratory Fitness Test, the impression is given that the protocol will be described, which is indeed the case, but the variables presented later in the Results section and in Table 4 are not described. The authors are kindly asked to address the identified shortcomings;
3.In section 3. Results, Table 1 describes "Physical Characteristics Variables and BMI of Participants" (as written in the title of the table), but the table itself also includes other variables such as Calcium Intake (mg/day), Calcium to Body Weight Ratio, cBPAQ, pBPAQ, and tBPAQ, which do not belong to the group of physical variables. Regarding Table 8, there is no discussion of the results within section 3. Results, unlike the previous seven tables. Additionally, it would be much clearer if each table was followed by its corresponding discussion, rather than the current format, where, for example, the accompanying text for Table 7 is in L209-212, and the table itself is in L255.

Author Response

This study explores the relationship between dietary calcium intake and muscle performance in young athletes. The study implies that improving dietary calcium intake might have broader benefits beyond bone health, particularly in athletic performance and overall musculoskeletal health, making it an important factor to consider in dietary interventions for athletes or individuals aiming to optimize physical performance.
This is well written article and the topic is of great significance. The research was conducted according to the principles and rules that apply to cross-sectional examination.
The methodology is thorough and facilitates an analysis of the research data.

However, the manuscript has the following deficiencies

  1. The title of the paper "Exploring The Role of Dietary Calcium Intake In Muscle Performance Among Young Athletes" is not aligned with the objective of the study (as mentioned in the abstract, L9-10, and similarly in the research goal stated in the Introduction section, L77-79), where it is stated, "This study aims to compare dietary calcium intake and its relationship with cardiovascular and muscular performance in young athletes (lacrosse, baseball, and soccer players)," which introduces the term "cardiovascular performance." Why is the term "Cardiovascular Performance" not included in the title, but only "Muscle Performance" is?;

We appreciate your feedback. This study focuses on the difference in dietary calcium intake across different sports. Interestingly, we found a negative correlation between dietary calcium intake and Vo2 and a positive correlation with the push-up test. We value your feedback and will incorporate the changes in the title. The new title is now “Exploring the role of dietary calcium intake in muscle and cardiovascular performance in young athletes”.

  1. In the section Materials and Methods, specifically subsection 2.2 Body Composition Variables, the testing protocol is described, but there is no mention of the variables.

Thank you for your feedback, we incorporated your suggestions as follow: Prior to the muscular performance test, each participant's height and weight were measured using a wall-mounted stadiometer (Novel Products, Rockton, IL, USA) and a digital scale bioimpedance analysis system (Tanita Inc., Arlington Heights, IL, USA), respectively. Furthermore, the variables such as %body fat, fat free mass, fat mass, and total body water were collected.

The same applies to subsection 2.3 Muscle Performance Variables, where the protocol is described, but the variables are not mentioned. Finally, in subsection 2.4 Cardiorespiratory Fitness Test, the impression is given that the protocol will be described, which is indeed the case, but the variables presented later in the Results section and in Table 4 are not described.  “

We added” The variables including time in the air, jump height, jump velocity, and power, were analyzed”. We also added “ During this test, the participant's resting and post-exercise hemodynamic responses (heart rate and blood pressure) were also measured.”

The authors are kindly asked to address the identified shortcomings;
3.In section 3. Results, Table 1 describes "Physical Characteristics Variables and BMI of Participants" (as written in the title of the table), but the table itself also includes other variables such as Calcium Intake (mg/day), Calcium to Body Weight Ratio, cBPAQ, pBPAQ, and tBPAQ, which do not belong to the group of physical variables.

Thank you for your feedback, we tried to summarize all these variables in one table, therefore we revised the title for Table 1.

Table 1. Physical Characteristics, BMI, Calcium Intake, and Bone-Specific Physical Activity Scores of Participants (Mean ± SD)

Variables

Baseball Players (n=49)

Soccer Players (n=33)

Lacrosse Players (n=13)

Age

19.89 ± 1.37

19.96 ± 1.59

19.46 ± 2.43

Height (cm)

180.10 ± 7.25

176.92 ± 7.90

177.80 ± 6.47

Weight* (lb)

83.37 ± 11.52

74.68 ± 9.32

82.11 ± 9.67

BMI*

25.71 ± 3.52

23.89 ± 3.06

25.91 ± 2.35

Calcium Intake (mg/day)*

1091.86 ± 437.91

928.64 ± 309.61

1321.25 ± 540.06

Calcium to body weight ratio

12.40 ± 5.32

12.70 ± 4.60

16.34 ± 6.75

cBPAQ**

7.77 ± 3.04

16.26 ± 14.93

10.75 ± 4.08

pBPAQ**

51.68 ± 21.08

49.12 ± 30.25

70.44 ± 34.24

tBPAQ*

29.86 ± 10.89

37.08 ± 18.79

40.59 ± 16.96

 BMI; Body Mass Index; cBPAQ; Current Bone-specific Physical Activity Questionnaire; pBPAQ; Past Bone-specific Physical Activity Questionnaire; tBPAQ; Total Bone-specific Physical Activity Questionnaire * significant p<0.05; ** significant p<0.01

 Regarding Table 8, there is no discussion of the results within section 3. Results, unlike the previous seven tables. Additionally, it would be much clearer if each table was followed by its corresponding discussion, rather than the current format, where, for example, the accompanying text for Table 7 is in L209-212, and the table itself is in L255.

Thank you for your feedback. We added commentary in the discussion section addressing the findings from table 8. Specifically, we added a discussion on LHG as an important predictor of muscle performance, which we had previously overlooked.

Reviewer 3 Report

Comments and Suggestions for Authors

The article provides interesting data on the role of calcium intake in muscle performance among young athletes. However, there are several areas that require significant development and improvement in order for the paper to have greater scientific value.

The introduction, while containing essential information about calcium and its importance for health, needs considerable expansion. There are many threads that should be included. First and foremost, it is important to highlight the diversity of micronutrients and their interactions in the context of sports performance. The authors should consider the impact of other minerals, such as magnesium and potassium, on muscle performance and the overall health of athletes. Additionally, it would be beneficial to include an overview of how the diets of young athletes may vary across different sports, as well as the specific nutritional needs associated with those sports.

Another important issue is the role of calcium in the context of other nutrients, especially vitamin D and protein, which influence calcium absorption and its functions in the body. The interactions of these components could be crucial for muscle performance. It is also worth describing the potential consequences of calcium deficiency on the health of young athletes, both in terms of physical performance and bone health. Introducing the topic of cultural differences in the diets of athletes could enrich the context of the study, as well as pointing to new research directions regarding the influence of calcium on performance could increase interest and engagement in the subject.

At the end of the introduction, clearly formulated research hypotheses and questions should be presented, which would allow for a better understanding of the study's objectives and facilitate reference to these issues in the later parts of the article.

In the context of research ethics, it is worth asking whether the authors, in addition to adhering to the Declaration of Helsinki, followed the guidelines outlined in the Publication Manual of the American Psychological Association regarding ethics and participant anonymity. If so, they should mention this in the article to enhance its credibility.

Analyzing the bibliography, I notice that only 7 out of 41 references relate to studies published in the last five years. This number is definitely insufficient to provide a solid foundation for the research. I urge the authors to include a greater number of recent publications so that at least 50% of the references come from the latest studies. This would significantly enhance the scientific value of the article and its relevance to current trends in research on calcium and muscle performance.

Author Response

The article provides interesting data on the role of calcium intake in muscle performance among young athletes. However, there are several areas that require significant development and improvement in order for the paper to have greater scientific value.

The introduction, while containing essential information about calcium and its importance for health, needs considerable expansion. There are many threads that should be included. First and foremost, it is important to highlight the diversity of micronutrients and their interactions in the context of sports performance. The authors should consider the impact of other minerals, such as magnesium and potassium, on muscle performance and the overall health of athletes. Additionally, it would be beneficial to include an overview of how the diets of young athletes may vary across different sports, as well as the specific nutritional needs associated with those sports.

Thank you for your feedback. We revised the following section:

MTEs such as calcium, magnesium, zinc, and selenium provide a significant role in physiological processes [1] correlated with sports performance, including energy storage/utilization, protein metabolism, inflammation, oxygen transport, cardiac rhythms, bone metabolism, and immune function [2,3].

It has been widely accepted that macronutrients play an important role in using muscle fuel and providing support to meet specific exercise demands [42]. It is worth noting that athletes' physical activity, performance, and micronutrient consumption must align with adequate standards [5]. Meeting the daily requirements seems crucial in addressing the deficiency related to specific MTEs among athletes for those who cannot meet their daily requirements. In this context, it is imperative to understand that athletes participating in competitive sports may encounter the challenges associated with injury risks necessitating prolonged recovery periods and adherence to a balanced diet. While no single dietary approach has been universally proven to optimize performance, emerging research has highlighted the Mediterranean diet as a model for enhancing athletic performance and overall well-being among athletes [6].

We have also included your suggestion in the discussion section:

Alternatively, these findings could be confounded by differences in in-season versus off-season behaviors across sports. Whereas soccer was in-season during data collection, baseball and lacrosse were off-season. Overall workload-to-recovery ratios during in-season and off-season training/practice probably vary within and between each sport. For example, evidence suggests that energy expenditures and physical activity levels differ between collegiate lacrosse players and basketball players during different team-based activities [40].

Another important issue is the role of calcium in the context of other nutrients, especially vitamin D and protein, which influence calcium absorption and its functions in the body. The interactions of these components could be crucial for muscle performance. It is also worth describing the potential consequences of calcium deficiency on the health of young athletes, both in terms of physical performance and bone health. Introducing the topic of cultural differences in the diets of athletes could enrich the context of the study, as well as pointing to new research directions regarding the influence of calcium on performance could increase interest and engagement in the subject.

We appreciate your valuable feedback. We revised with the following information in the introduction section:

Irrefutable evidence supports vitamin D regulates calcium absorption and mineralization while maintaining muscle mass, muscle strength, and overall physical performance. This is particularly evident in older individuals, highlighting the role of vitamin D and calcium in muscle performance and mitigating sarcopenia [13], especially in older people, suggesting its importance in muscle performance.  However, no direct evidence supports calcium supplementation's efficacy in athletic performance [1]. It is worth noting that calcium's role in muscle contraction can mitigate the impact of increased levels of parathyroid hormones, which stimulate bone resorption [14] and increase the risk of osteoporosis. In this sense, it is important to acknowledge that a research gap exists in exploring the role of dietary calcium intake and muscle performance in young athletes.

Although we did not collect information regarding the cultural differences in the participant's diets, we included this in future research suggestions.

It is important to note that dietary planning plays a crucial role in determining athletes' physical and cognitive performance, yet micronutrients are often overlooked compared to macronutrients in their diet. Poor micronutrient composition could possibly be due to a lack of education, while sports nutrition knowledge has significantly proven appropriate food selection among athletes. Therefore, future research suggestions should incorporate wide-ranging sports teams along with intervention studies in male and female athletes and further expand research on cultural differences in diet and pattern of diet along with intervention studies

At the end of the introduction, clearly formulated research hypotheses and questions should be presented, which would allow for a better understanding of the study's objectives and facilitate reference to these issues in the later parts of the article.

We appreciate your feedback. We revised this section and added, the following information:

Although the importance of dietary calcium intake has been well studied in bone metabolism, there is still a wide research gap in understanding the relationship between calcium intake and muscle performance, especially in young athletes. Therefore, the present study aims to compare dietary calcium intake across different sports and its relationship with cardiovascular and muscular performance in young athletes. We hypothesized that there would be a variation in dietary intake between soccer players, lacrosse players, and baseball players and that dietary calcium levels would be associated with cardiovascular and muscle performance.

In the context of research ethics, it is worth asking whether the authors, in addition to adhering to the Declaration of Helsinki, followed the guidelines outlined in the Publication Manual of the American Psychological Association regarding ethics and participant anonymity. If so, they should mention this in the article to enhance its credibility.

Thank you for your suggestion.

Adherence to both the Declaration of Helsinki and the ethical guidelines from the APA's Publication Manual would certainly enhance the article's credibility. We included the Declaration of Helsinki and the ethical guidelines according to the journal guidelines.

Analyzing the bibliography, I noticed that only 7 out of 41 references relate to studies published in the last five years. This number is definitely insufficient to provide a solid foundation for the research. I urge the authors to include a greater number of recent publications so that at least 50% of the references come from the latest studies. This would significantly enhance the scientific value of the article and its relevance to current trends in research on calcium and muscle performance.

 Thank you for your suggestion. We tried to include the most relevant articles on this topic, and now we have included 17 articles published after 2019.

We appreciate your suggestions.

Round 2

Reviewer 1 Report

Comments and Suggestions for Authors

Dear Authors,

thank you very much for making corrections and corrections as well as supplementing the manuscript with the content I asked for. The quality of the work has significantly improved, so I recommend it for publication

Reviewer 3 Report

Comments and Suggestions for Authors

Thank you very much for taking my reviews into account. The current version of the article is significantly improved, and I have no further comments.